# Analysis of Causes of Porosity Change of Castings under the Influence of Variable Biscuit Height in the Filling Chamber

**DOI:** 10.3390/ma14226827

**Published:** 2021-11-12

**Authors:** Štefan Gašpár, Ján Majerník, Jan Kolínský

**Affiliations:** 1Department of Technical Systems Design and Monitoring, Faculty of Manufacturing Technologies with a Seat in Prešov, Technical University of Košice, Štúrova 31, 080 01 Prešov, Slovakia; stefan.gaspar@tuke.sk; 2Department of Mechanical Engineering, Faculty of Technology, Institute of Technology and Business in České Budějovice, Okružní 517/10, 370 01 České Budějovice, Czech Republic; kolinsky@mail.vstecb.cz

**Keywords:** HPDC, porosity, air entrapment, aluminium alloys, gating system

## Abstract

Quality properties of castings produced in a die casting process correlate with porosity that is conditioned by a number of factors, which range from input melt quality to setup of technological factors of the die casting, and through structural design of the gating system. One of the primary parameters conditioning the inner soundness of the casting is the liquid metal dose per single operation of die casting. This paper examines the issue of metal dose. The experiments are performed with casting a gate system of an electromotor flange. The gating system examined was die cast with a variable volume of metal dose per single operation. The metal dose was adjusted to reach the height of a biscuit of 10, 20, and 30 mm. The examination of the inner homogeneity of the castings of the individual variants of gating systems with variable height of the biscuit proved that decreasing biscuit height results in an increase of porosity share in the casting volume. The programme MagmaSoft 5.4 revealed the main causes of changes in porosity share. The simulations detected that the change in biscuit height and volume of liquid metal directly influence thermal conditions of the melt in the filling chamber, and in the mould by means of the period in which the holding pressure action is influenced. Simultaneously, the melt flow mode in the sprues and gas entrapment in the melt volume are affected as well. Correlation of the factors consequently influences the final porosity of castings.

## 1. Introduction

Technology of high pressure die casting (HPDC) represents a special type of die casting method in modern technology of metal processing, which requires minimum or even null finishing operations in the case of parts production. Owing to its high productivity, HPDC is widespread in different industrial sectors [1]. Casts produced in the die casting process are characterized by high geometrical precision, good mechanical properties, and low price. Mechanical properties of casts are closely related to formation of fine-grained structure in the case of fast melt cooling with the mould face [2]; however, defects such as porosity, primarily caused by air entrapment by the melt during the filling phase, significantly influence the quality of casts [3].

Due to increased porosity occurrence and different share of air entrapment in the cavities, which occur as the result of shrinkage or of gas entrapment in the mould cavity, the porosity type identification in the HPDC casts appears to be complicated. In general, occurrence of porosity in the internal structure of casts is ascribed to change in the state of matter (liquid/solid matter). The cavities, with size ranging from microns to millimetres, depend on the type of metal alloy and on solidification process of casts. Formation of cavities in the structure of casts originates in the melt shrinkage during solidification, in distribution of gasses present directly in the melt, or in combination of these factors, including gaseous porosity caused by gas entrapment in the course of the cycle of high pressure die casting (varying plunger speed, venting, and gating system design) [4,5,6].

To produce a high quality and reliable cast during the HPDC process, synchronization of several process parameters is required. Therefore, the HPDC process must be considered as a reciprocally correlating mechanism of structure of a die casting machine, mould designs, and setup of technological parameters of the casting process [7,8]. Therefore, it is important to focus on the issue of reduction of porosity formation in the volume of casts partly in the individual design phases and to find the most adequate intersection of the individual solutions [9,10].

Currently, the factor influencing the reduction and compression of pores most discussed is holding pressure. It has been proved that increased values of holding pressure positively affect the distribution and size of pores by means of which they influence the tightness of pores. After solidification of gates, the volume of pores slightly increases, which is caused by the melt shrinkage during solidification. On the other hand, increase of holding pressure shortens the service life of a mould [7,11,12].

Out of different process variables, a significant factor influencing gas entrapment in the volume of the cast is the melt speed in the gate. The speed directly determines the filling mode of the mould-shaping cavity. The melt speed in the gate is directly proportional to the speed of the pressing plunger in the filling chamber. Higher pressing speed changes the character of the flow in the runner from laminar–planar to turbulent–non-planar, which causes discontinuous melt flow. By means of the pressing speed reduction, the melt flow can be tranquillized, which results in a continuous and regular face of the melt flow in the cross-section of the runner. On the other hand, extreme prolongation of the casting cycle at low pressing speed leads to a decrease of the melt temperature, which may result in occurrence of defects such as incomplete topping up, cold laps, and weld lines [13,14,15,16].

Apart from establishing the technological parameters of the die casting, a crucial factor assuring the quality of castings is the design of the gating system [3]. In general, the design of the gating system should allow for fast filling of the mould-shaping cavity by the liquid melt so that the metal flows through the mould cavity along straight trajectories without abrupt changes in the melt flow at wide angles [17,18,19,20]. Air entrapment by the melt during the filling phase represents one of the basic reasons for porosity of casts. Correct design of the gating system can influence the filling mode of the mould cavity filling, which is influenced by a number of factors including shape and massiveness of the cast, weight ratio between the cast and the mould, gating system disposition, shape and area of the gate, mould cavity volume, and the area and arrangement of venting holes. It has been proved that approximately 90% of defects of casts produced in the die casting process are caused by the errors in the gating system design [16,21,22,23].

A rather significant yet often underestimated factor in the case of die casting of metals is correctly setting the batch weight/volume of material required for a single operation. The weight is determined by a sum of weights of a clean casting in a single-cavity mould and, in case the weight is given by weight of all casts, by weight of the fins, runners, and residues in the filling chamber. In total, it is weight/volume of metal poured into a shot chamber per single operation [24].

Utilization of a batch is given by the ratio of the sheer weight of the cast and batch weight, which is usually 25–30% in the case of small and medium size casts and 50–60% in the case of heavy casts, reaching even 80% in the case of very heavy casts. The metal residues in a filling chamber have the greatest weight share in the batch weight. As the batch weight significantly influences the price of the cast, the returnable material such as runners, fins, and residues in the chamber must be reduced to a minimum amount technologically required for production of high-quality casts [25].

In the case of a horizontal filling chamber, the die casting ratios are more advantageous than in the vertical case. If the pressing plunger reaches the mould joint, the biscuit height in the filling chamber can be minimal, provided that the metal dose per single operation is correct [26,27].

This paper deals with the issue of how height influences the porosity of castings in a runner residue in the filling plunger. The sets of casts were produced with variable biscuit height Z in the case when the biscuit height was selected with values of Z1 = 10, Z2 = 20, and Z3 = 30 mm. Based on evaluation, it was detected that increasing height of biscuit Z causes decreasing values of cast porosity. Consequently, simulations of the pouring cycle were conducted by means of the programme MagmaSoft 5.4 with the variable height of biscuit Z, which were used to identify the main causes of changes in porosity share in casts. The monitored values were temperature of the melt in the filling chamber, temperature of the melt in the area of the gates, and the mode of the melt flow in the runners. Based on simulations, it was proved that increasing height of biscuit Z, which is closely connected to batch volume, resulted in an increase of the melt temperature in the areas of the measuring spots. This also influences the period of the melt solidification in the area of the gates, which directly conditions the period duration of the holding pressure effect and pore reduction in the cast volume. Concurrently, with the lower height of the biscuits, the ratio of the metal and gas volume in the gating system changes as well, by means of which the mode of the melt flow in the runners is affected. Correlation of these factors, i.e., lower temperature of the melt under the concurrent change of the mode of melt flow, results in an increase of porosity share in the volume of casts with a decreasing biscuit height Z.

## 2. Materials and Methods

Evaluation of biscuit height influence on the porosity values of casts and consequent cause analysis of changes were conducted with the cast of the electromotor flange with the relevant gating system (Figure 1).

Porosity share in the castings was examined at the points in which further mechanical machining of the casts is performed (Figure 1—porosity monitoring locations). The points were evaluated as crucial to the possibility of revealing cavities in the cast volume in machining of the holes.

The influence of the biscuit height on porosity share was monitored. The biscuit height was conditioned by batch volume or by metal dose volume per single operation. Table 1 shows the basic volume characteristics of the gating system together with variable values of batch volume and biscuit height Z.

To assure relevant results that demonstrate the influence of biscuit height on porosity values in the monitored areas, the individual series of casts were die cast with a consistent setup of technological parameters of the die casting cycle. The values of technological parameters used in the setup are shown in Table 2.

Inner homogeneity of the analysed casts was primarily assessed by means of a nondestructive method in the RTG laboratory through evaluation of the images generated by RTG VX1000D (North Star Imaging, Marlborough, Marlborough, MA, USA) equipment. Consequently, the share of porosity in the selected points was evaluated as percentage share of pores in the scratch pattern of the monitored point. Porosity analysis of scratch patterns of the specimens was performed using an OLYMPUS GX51 (Tokyo, Japan) microscope at 100× consequently, the analysis was processed by the programme ImageJ.

Possible causes of change of porosity share in casts with regard to change of biscuit height were examined by means of the programme Magmasoft MAGMA 5.4.1—HPDC module. Setup of input parameters for the simulation was identical to the setup of technological parameters of the die casting cycle for the case in which die casting of testing casts was performed (Table 2). To improve accuracy of simulation and to obtain a more detailed description of the target entity, a grid with high fineness and efficiency of generation was formed. In the case of the die casting simulations, a fine grid with 89,464,224 cells was used. However, the gating system consisted of 1,671,742 cells.

To clarify the causes of changing porosity under the influence of different biscuit heights, it was necessary to establish the following assumptions:To a high degree, the size and distribution of pores in casts are influenced by holding pressure. Its influence is conditioned by the action of hydrostatic pressure on the mould-shaping cavity. The action of holding pressure upon the melt in the mould-shaping cavity stops at the moment solidification of the melt is finished in the area of the gates, which is closely connected with its initial temperature when entering the particular point. The following opinion was expressed: the lower dose volume of liquid metal per single operation has lower thermal capacity and the melt has a tendency to premature solidification when passing through the gating system, thus reducing the period of holding pressure acting upon the melt in the mould-shaping cavity.The change of the metal volume dose per single operation leads to a change of the melt/gas ratio in the gating system. The filling time of the mould cavity filling was set to 16.14 ms, which implies that, with the biscuit height Z1 = 10 mm and within the same time interval, a higher volume of gasses and vapours are inevitably forced out than in the case of the biscuit with a height of Z3 = 30 mm. In such a case, a more striking mixture of melt and gas may occur, which directly increases porosity share in the cast volume.

Verification of the premise (a) was conducted by the application of module HPDC, Results—Filling/Temperature, for the case in which the monitored temperature of the melt in the filling chamber and in the area of the gate is at the points shown in Figure 1. To determine the solidification period of the melt in the area of the gate and to determine the period of holding pressure action, we used the HPDC module: Results/Solidification and Cooling until Eject/Fraction Solid. Assessment of the melt flow mode in the gates, verification of the premise (b), was conducted by applying the Results —Filling/Temperature module in HPDC concurrently with the evaluation of thermal conditions in the melt.

## 3. Results

The results obtained through experiments can be divided into three parts. The first part describes the change of porosity in the casts depending on the biscuit height. Consequently, the simulations are conducted which serve for analysis of the changes. Thermal conditions in the melt, length of holding pressure action, and mode of the melt flow through the sprues are examined.

### 3.1. Evaluation of Porosity of Castings

Porosity of casts was considered in the area of structural holes, which were evaluated as the risky points of the casts. RTG analysis was used to detect heterogeneity at the critical points. Consequently, the scratch patterns of the monitored points were produced in the area in which porosity was evaluated. Figure 2 presents a specimen of the group of casts characterized by the lowest porosity share. Figure 3 presents the specimen of the group of casts characterized by the highest porosity share.

The graph shown in Figure 4 presents the change in porosity of the castings depending on the height of the biscuit. On the left side of the graph, the porosity values of individual samples examined are presented. The right side presents the change in the average porosity of the castings depending on the change in the height of the biscuit.

### 3.2. Evaluation of Thermal Conditions of the Melt

When analysing the causes for the change of porosity share in the volume of casts, attention was primarily given to examination of thermal conditions of the melt during the pouring cycle. The temperature was monitored at the measuring points as shown in Figure 1. Three measuring points were placed in front of connection point of the filling chamber to the main runner. The temperature was monitored even in the gate area. Table 3 shows the measured values of melt temperature dependence on variable biscuit height. Temperature development was monitored with regard to plunger position in the filling chamber during the period while the melt filled the entire cross-section of the filling chamber. The melt temperature during the entrance of the melt into the measuring point (Filling Temperature) was monitored as well.

Table 3 unambiguously shows that decreasing biscuit height and with decreasing volume of the metal dose per single operation, the melt is extremely undercooled when passing through the filling chamber, which results in faster reduction of thermal capacity.

Based on Table 3, it is clear that the melt temperature in the area of the biscuit decreases depending on both plunger position and biscuit height. If the temperature difference of the melt in the filling chamber becomes visible at variable biscuit height, it is expected that temperature difference is measurable in the area of the gates as well. The temperature in the gate area was monitored at the measuring points, as shown in Figure 1. The temperature in the gate area was monitored during the entrance of the melt into the measuring point (Filling Temperature) and at the end of the filling phase, i.e., before the holding pressure action began.

Filling temperature also provides information on the melt temperature during melt entrance at the measuring point. Temperature at the end of the filling phase is the information on the melt temperature at the measuring point at the moment of complete filling of the mould cavity and prior to commencement of solidification phase. Based on the results of the melt temperature measurements in the gate, presented in Table 4, it is clear that for biscuit Z1 and Z2, the thermal potential of the melt is reduced. When passing through the runners, the melt, contrary to biscuit height Z3, is more undercooled. In the volume of metal dose per single operation at biscuit height of Z3, the amount of heat in the melt can compensate for the losses occurring through the runners. Therefore, the measuring points in the gate area and mould-shaping cavity are supplied with “fresh melt” and characterized by higher temperature, which positively influences porosity elimination.

Higher melt temperature in the area of the gates presupposes a longer period of holding pressure action upon the melt. The period of holding pressure action upon the melt in the area of the mould-shaping cavity was monitored by means of the HPDC module: Results/Solidification and Cooling until Eject/Fraction Solid. The period of time during which 100% of the solid particles in the area of gate was under monitoring occurred by means of which the transfer of hydrostatic pressure to the area of casts was eliminated. Table 5 presents duration period of holding pressure action for the individual variants of biscuit height.

According to Table 5, it is clear that increasing biscuit height Z during the period of holding pressure action upon the melt in the mould-shaping cavity is prolonged as well. Concerning the measured values of porosity of castings shown in Figure 4, it is clear that higher temperature of the melt used for filling the mould-shaping cavity together with a longer period of holding pressure action positively influence the elimination of porosity in the casts.

### 3.3. Analysis of the Melt Flow in the Runners

Based on the data shown in Table 2, it is clear that increasing biscuit height results in dosing a higher metal volume to the filling chamber of the machine per single operation. Increasing metal volume leads to decreased air and gas volume, which must be forced out from the gating system during the pouring cycle. Table 6 shows the volume characteristics of gasses present in the individual gating systems and their dependence on the biscuit height.

Generally, a smooth mould cavity filling is desired, without any mix of the melt and gasses. It means that gasses present in the gating system should be forced out gradually in front of the melt flow face. The same is also applicable in the case of filling the mould-shaping cavity when the filling along straight trajectories is required with gradual elimination of venting holes. Duration of filling time was set to 16.14 ms for all variants of the biscuit height. In the course of the interval, a higher volume of gasses must be forced out at biscuit height Z1 than at height Z3. Figure 5 and Figure 6 present development of the mould cavity filling for the individual variants of biscuit height.

Based on Figure 5 and Figure 6, it is clear that prior to completion of the filling phase with biscuit height of Z1 = 10 mm in the sprue, the gas is still present, which is consequently distributed to the volume of the cast. This fact explains the remarkable increase of cast porosity in the case of the Z1 series castings, contrary to casts Z2 and Z3, according to Figure 4.

## 4. Discussion

The experiments conducted proved the remarkable influence of biscuit height and liquid metal dose per single operation on cast porosity. It was proved that decreasing biscuit height leads to an increase of porosity in the casts. The aim of this paper was not only to describe dependence of the change of porosity of casts on biscuit height, but to clarify the causes of these changes as well.

The first cause of change in cast porosity is the thermal capacity of the metal comprised in the volume of dose per single operation. It was proved (Table 3) that in the case of a higher volume of metal dose per single operation with a constant pouring temperature of 705 °C, the melt flowing in the filling chamber cooled more slowly. This fact is related to the thermal capacity of the melt volume in the case in which the higher metal volume has, logically, higher thermal capacity.

Insufficient cooling of the melt in the filling chamber correlates with the melt temperature in the area of the gates. Based on Table 4, it is evident that in the case of a dosing volume of 481.05 cm^3^, corresponding to the biscuit height Z3, the amount of heat in the melt volume can compensate for heat loss occurring when the melt passes through the gating system. Due to this aspect, the measuring point—the area of the gate and mould-shaping cavity—are supplied with the “fresh melt“ at a higher temperature, which positively influences porosity elimination.

These observations indicate that the action period of holding pressure is prolonged, which results in diminishing size and distribution of pores in the cast.

A significant factor is also the volume of gases and vapours in the gating system prior to triggering of the pouring cycle. According to Table 6, the volume of gases in the gating system at biscuit height Z1 is 1307.82 cm^3^, which is 76.94 cm^3^ more than the 1230.89 cm^3^ volume for biscuit height Z3. The volume difference must be forced out of the mould cavity during the same period of time, i.e., 16.14 ms. A more massive mixture of gasses and melt occurs when the melt passes through the gating system, which is shown in Figure 5. In this situation, with biscuit height Z1 and with the overall filled volume of the gating system reaching 94%, the occurrence of gas in the runners can be observed during concurrent filling of the cast. The gas is consequently transported into the volume of the cast, as shown in Figure 6, which also explains the large increase of porosity of the casts produced at biscuit height Z1, contrary to casts with biscuit height Z2 and Z3. In simple words, the volume of gasses contained in the gating system with the biscuit height Z1 is incapable of release at the melt flow face, and is therefore mixed with liquid metal.

Thus, it can be concluded that porosity occurrence in casts produced in the die casting process is strongly influenced by the following:Thermal capacity of the melt volume, which conditions the cooling speed of the melt;Melt temperature, which influences the phase of holding pressure action;Ratio of melt and gasses in the gating system of the mould, which conditions dragging of gasses by the melt and their mutual mixture;Mutual correlation of the aforementioned factors.

## 5. Conclusions

The average porosity values measured in the experiment are presented in Figure 3 and are characterized by noticeable differences in the dependence on height of the metal residue (of the biscuit) in the filling chamber. The results proved that in the case of all specimens die cast at biscuit height Z1 = 10 mm (specimens A1–A5), the porosity share appears to reach its highest occurrence. Remarkable occurrence of porosity in the case of the respective specimens is influenced by the amount of liquid metal at the expense of increased air volume in the filling chamber. In the course of the mould cavity filling, the air is dragged off the chamber by the liquid metal to the cast. At low biscuit height, holding pressure becomes evident only with certain difficulties. However, increased biscuit height in the filling chamber to the values of Z2 = 20 mm and Z3 = 30 mm proved to have positive influence on the reduction of porosity occurrence. Therefore, each type of cast produced in the die casting process requires determination of the optimal average of the filling chamber, together with the volume of liquid metal volume required for the single pouring operation.

## Figures and Tables

**Figure 1 materials-14-06827-f001:**
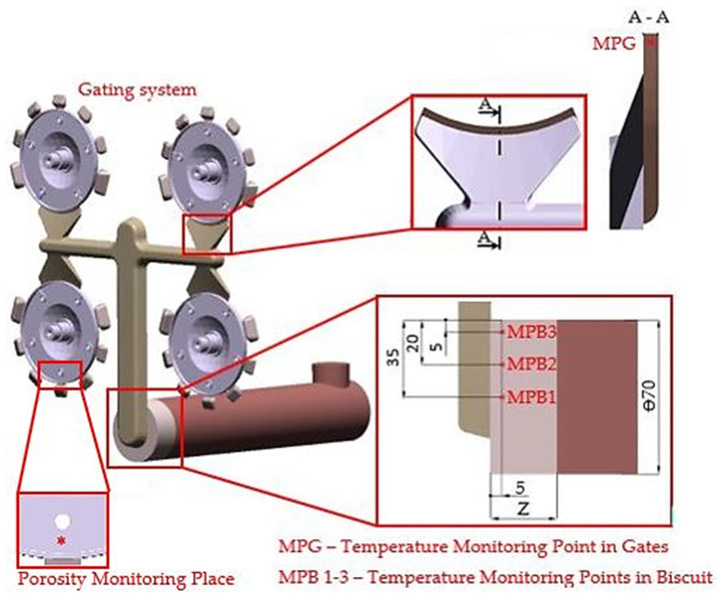
Location of measuring points.

**Figure 2 materials-14-06827-f002:**
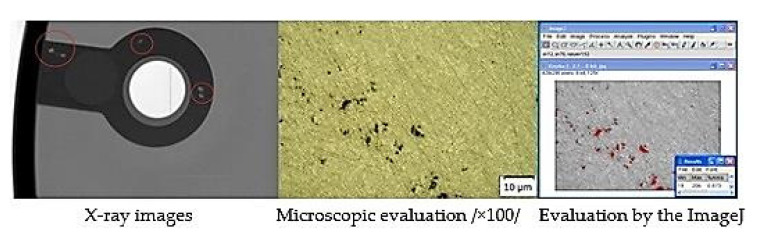
Analysis of specimen C.1.

**Figure 3 materials-14-06827-f003:**
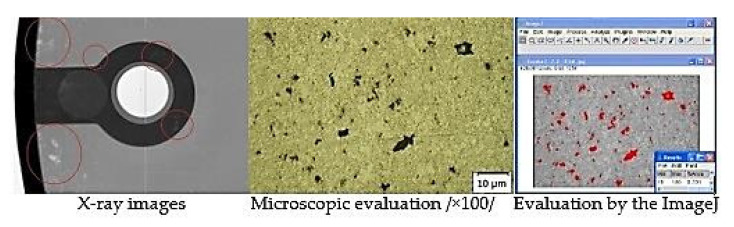
Analysis of specimen A.1.

**Figure 4 materials-14-06827-f004:**
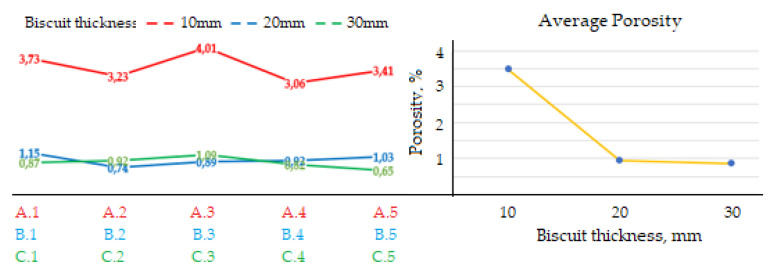
Porosity change dependence on biscuit height.

**Figure 5 materials-14-06827-f005:**
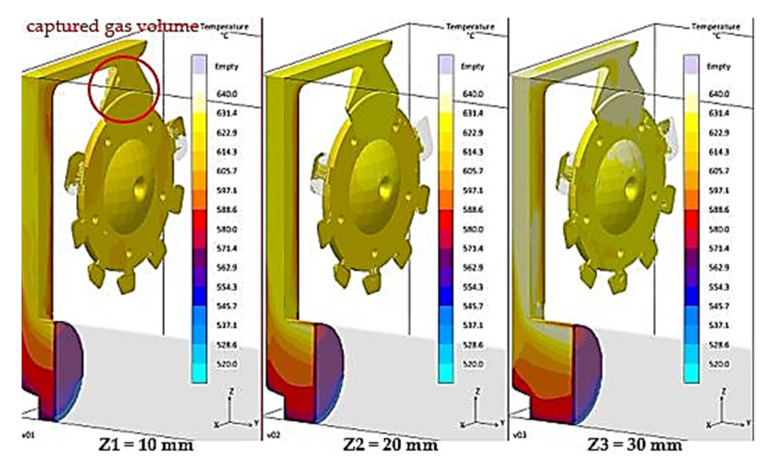
Gas entrapment in the case of 94% filling volume.

**Figure 6 materials-14-06827-f006:**
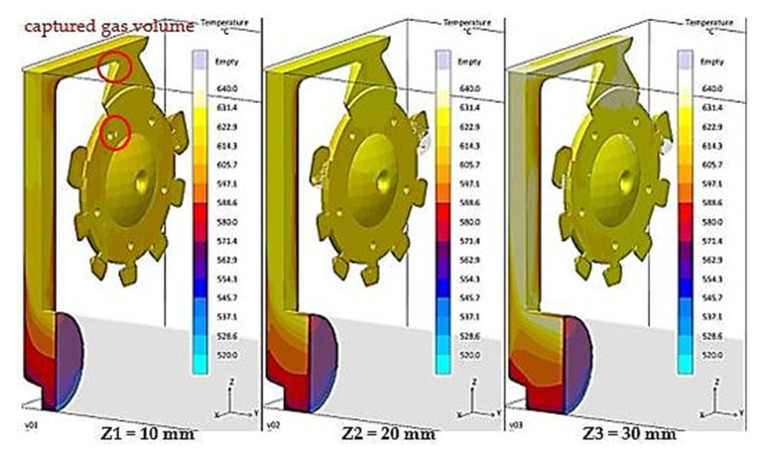
Gas entrapment in the case of 97% filling volume.

**Table 1 materials-14-06827-t001:** Characteristics of the gating system.

Shot Chamber Properties			
**Property**			**Value**
Shot chamber length, mm			350
Plunger position when pour hole is covered, mm			49
Plunger diameter, mm			70
**Volumes**			**Value**
Runner, cm^3^			207.25
Cavity without Biscuit and Runner, cm^3^			158.41
Shot chamber volume, cm^3^			1346.28
**Biscuit**			
Biscuit height, mm	**Z1**	**Z2**	**Z3**
	10	20	30
Biscuit volume, cm^3^	38.46	76.93	115.39
Dosing volume, cm^3^	404.12	442.59	481.05
Filling the chamber, %	30.01	32.87	35.73

**Table 2 materials-14-06827-t002:** Technological parameters of casting cycle.

Technological Parameters of Casting Cycle	
Parameter	Value
Alloy	EN-AC 47 100 (AlSi12Cu1(Fe))
Pouring temperature, °C	705
Die temperature, °C	200
Plunger velocity 1st phase, m/s	0.8
Plunger velocity 2nd phase, m/s	2.6
Intensification pressure, MPa	25

**Table 3 materials-14-06827-t003:** Melt temperature in the area of the biscuit and its dependence on biscuit height Z.

**Biscuit Height 10 mm**				
	**Temperature, °C**			
**Piston Position, mm**	**MPB1**	**MPB2**	**MPB3**	**Average**
250	633.3	623.6	600.6	619.17
260	631.3	616.5	595.9	614.57
270	630.3	611.2	594.4	611.97
280	629.6	607.3	594.6	610.50
290	628.3	605.3	595.3	609.63
300	625.9	602.8	595.3	608.00
310	623.5	600.4	595.54	605.80
320	621.3	598.2	591.3	603.60
330	619.1	596.0	589.2	601.43
340	616.9	593.8	586.9	599.20
Filling Temperature, °C	633.4	627.3	594.6	618.43
**Biscuit height 20 mm**				
	**Temperature, °C**			
**Piston position, mm**	**MPB1**	**MPB2**	**MPB3**	**Average**
240	640.5	629.1	605.3	624.97
250	627.4	622.1	601.1	616.87
260	636.5	615.9	598.3	616.90
270	636.2	611.9	597.5	615.20
280	635.6	609.5	597.8	614.30
290	634.5	607.9	598.1	613.50
300	632.2	605.7	597.1	611.67
310	630.0	603.5	595.3	609.60
320	627.6	601.1	592.9	607.20
330	625.7	599.3	591.0	605.33
Filling Temperature, °C	640.4	631.1	602.9	625.13
**Biscuit height 30 mm**				
	**Temperature, °C**			
**Piston position, mm**	**MPB1**	**MPB2**	**MPB3**	**Average**
230	648.7	642.1	611.4	634.07
240	644.4	634.0	608.7	629.03
250	643.1	626.5	605.4	625.00
260	642.8	619.8	602.4	621.67
270	642.5	615.0	601.0	619.50
280	642.2	612.7	600.6	618.50
290	640.6	611.0	600.0	617.20
300	638.4	608.8	598.6	615.27
310	636.0	606.5	596.5	613.00
320	634.3	604.7	594.8	611.27
Filling Temperature, °C	649.8	644.6	614.3	636.23

**Table 4 materials-14-06827-t004:** Values of temperature of the melt in the gate area and its dependence on biscuit height Z.

**Biscuit Height 10 mm**					
	**Temperature, °C**				
	**MPG1**	**MPG2**	**MPG3**	**MPG4**	**Average**
Filling Temperature, °C	619.9	620.3	623.4	620.0	620.90
End of Filling, °C	619.8	620.2	619.9	619.6	619.88
**Biscuit height 20 mm**					
	**Temperature, °C**				
	**MPG1**	**MPG2**	**MPG3**	**MPG4**	**Average**
Filling Temperature, °C	625.1	625.9	629.1	629.1	627.30
End of Filling, °C	626.8	626.3	626.6	626.0	626.43
**Biscuit height 30 mm**					
	**Temperature, °C**				
	**MPG1**	**MPG2**	**MPG3**	**MPG4**	**Average**
Filling Temperature, °C	630.5	629.8	631.2	629.1	630.15
End of Filling, °C	634.0	633.9	633.5	633.5	633.74

**Table 5 materials-14-06827-t005:** Duration period of holding pressure action and its dependence on biscuit height.

Biscuit Height	Z1 = 10 mm	Z2 = 20 mm	Z3 = 30 mm
Intensification pressure duration	457.6 ms	463.4 ms	468.5 ms

**Table 6 materials-14-06827-t006:** Volume of gas and melt and its dependence on biscuit height.

Biscuit Height	Z1	Z2	Z3
Volume of filling chamber + die cavity, cm^3^	1711.94		
Dosing volume, cm^3^	404.12	442.59	481.05
Gas volume, cm^3^	1307.82	1269.35	1230.89

## Data Availability

Data is contained within the article.

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
