# Peer review of "Analysis of Causes of Porosity Change of Castings under the Influence of Variable Biscuit Height in the Filling Chamber"

_materials, 2021, doi:10.3390/ma14226827_

Round 1

Reviewer 1 Report

The submitted manuscript focuses on the role of the biscuit height in influencing the porosity in the final cast product. I think the topic is of interest and deal with a practical aspect of the HPDC process.

I would recommend changing the data presentation: Table 3 and Figure 4 report the same data so there is a redundancy. Besides, standard deviation/error or error bars are necessary in order to read correctly the numbers, in either the table or the figure. In the case of tables, the "average ± SD/SE" enables the authors to avoid listing the entire data set.

The same comment holds for Table 4 and Figure 5, and Table 5 and Figure 6: either the table or the figure should be presented, not both, and with appropriate SD or SE.

I would like to suggest spell check in the text, for example:

line 13 "pf" should be "of"

line 34 "spheres of industry" should be "industrial sectors"

line 43 "structure of cast" the meaning is not clear

line 91 "use of a batch is given" the meaning is not clear

line 99 "the pressing plunger reaches the dividing plane of the mould" should be rephrased

line 279 "volume o fair" should be "of air"

Author Response

Thank you for your review and comments on improving the manuscript

All your comments have been accepted and edited in the manuscript

The original Table 3 has been deleted

The original Figure 5 and Figure 6 has been deleted

Spelling mistakes, typos and wording have been fixed (highlighted in red)

Reviewer 2 Report

English is rather OK. It is not hard to understand what authors meant, but minor English language correction (spelling and style) is suggested – as an example:

  • "results in increase of share of porosity in the volume of casts" should be rewritten to sound less like taken directly form the translator i.e. : results in increase of porosity share in the volume of casts”
  • “the ration of the metal volume and gas volume” should be rather: the ratio of the metal and gas volume”
  • “Evaluation of influence of the biscuit height on the porosity values of casts and consequent analysis of cause of the changes were conducted” should be more like this: Evaluation of biscuit height influence on the porosity values of casts and consequent cause analysis of changes were conducted
  • “chippy machining of the holes” – I have no idea wat chippy machining is

Examples of errors found in the text:

  • Table 1: missing of diameter value unit (probably mm)
  • Table 2: please change alloy designation from EN AC 47 100 to EN AC-47100
  • Table 2: I would change the style of the velocity unit from “m.s-1” to m/s
  • Table 3: in the text describing the table Authors mention the height but in the table there is a thickness
  • 4: same as above + I had problems understanding the graph on the left – its has no axis units and no axis units labels. In my opinion this graph on the left is unnecessary – table above shows the values in a sufficient way.

General remarks to the article:

  • A more detailed description of FEM analysis is needed in my opinion. There are no boundary conditions of FEM simulations described besides of finite elements numbers. It would be also a good practice to clearly indicate which results where obtained through FEM simulations (i.e. simply by clarifying it in the text or by adding a remark such as (FEM results) at the end of the table description.
  • In many cases height of the biscuit is mistaken with the thickness of the biscuit. It would be recommended to unify this property along the text and figures/tables.
  • I don’t understand why authors use dashed line in the legend of the charts while on the graph itself they only use a solid line style. It would be recommended to unify that.

Author Response

Thank you for your review and comments on improving the manuscript

All your comments have been accepted and edited in the manuscript

Spelling mistakes, typos and wording have been fixed (highlighted in red)

Reviewer 3 Report

After reading the article, please complete or answer the following issues:

  1. Table 1, Plunger diameter, no unit.
  2. Table 1, please replace commas with dots.
  3. Page 6, can show table 3 first, and below fig. 3 ?, then you know what C1 and A1. And transfer lines 202, 203 to line 197.
  4. Line 202, ……………table 3 to one line.
  5. Lines 222 and 223 of the table caption should be divided into two lines or merged into one line.
  6. Please move the text so that the table 4 was on one page. The same applies to table 5.
  7. Line 314, is the correct marking in table 5 and figure 6?
  8.  Line 343, redundant period at the end of the sentence.

All drawings should be of better quality, maybe just paste the original drawings (JPG) into the text?

Author Response

Thank you for your review and comments on improving the manuscript

All your comments have been accepted and edited in the manuscript

Spelling mistakes, typos and wording have been fixed (highlighted in red)

In view of another reviewer's comment, the original Table 3 and Figures 5 and 6 have been removed, so some comments on text formatting are resolved